# Unveiling the global influence of tropical cyclones on extreme waves approaching coastal areas

**Swen Jullien** [1] ✉, **Jérôme Aucan** [2], **Elodie Kestenare**[3], **Matthieu Lengaigne** [4] **& Christophe Menkes** [2]

Tropical and extra-tropical storms generate extreme waves, impacting both nearby and remote regions through swell propagation. Despite their devastating effects in tropical areas, the contribution of tropical cyclones (TCs) to global wave-induced coastal risk remains unknown. Here, we enable a quantitative assessment of TC's role in extreme waves approaching global coastlines, by designing twin oceanic wave simulations with and without realistic TC wind forcing. We find that TCs substantially contribute to extreme breaking heights in tropical regions (35-50% on average), reaching 100% in high-density TC areas like the North Pacific. TCs also impact remote TC-free regions, such as the equatorial Pacific experiencing in average 30% of its extreme wave events due to TCs. Interannual variability amplifies TC-induced wave hazards, notably during El Niño in the Central Pacific, and La Niña in the South China Sea, Caribbean Arc, and South Indian Ocean coastlines. This research offers critical insights for global risk management and preparedness.

Extreme waves and storm surges driven by both extra-tropical and tropical storms are primary factors contributing to extreme coastal water levels. In low-lying coastal areas, floods induced by tropical cyclones (TCs) account for over 90% of human and property losses[1]. TCs are therefore recognized to be among the most devastating natural hazards globally[2]. With the looming threat of climate change, marked by accelerated sea-level rise, intensification of the most extreme events, and diminished coastal resilience, hazards and impacts posed by cyclones are expected to further exacerbate[3–6]. It is becoming increasingly urgent to provide a comprehensive quantitative assessment of TC-related coastal risk.

However, two significant gaps are hindering such assessment. Firstly, existing global models lack the requisite spatial resolution to accurately represent TC wind fields[7–10]. Secondly, the contribution of waves to extreme water level, i.e. the wave setup, is often overlooked[2,3,11–13]. This stems from the fact that coastal observations are primarily obtained from tide gauges, which, often located in wave-sheltered areas like ports, do not allow for an accurate estimation of

wave setup. Past studies estimating extreme water levels at the coasts have thus relied on empirical formulations of the wave contribution, drawing from offshore wave heights, wavelengths, and foreshore slopes (see ref. 14 for a review). They suggested that waves could significantly contribute to extreme coastal water levels, with their contribution potentially exceeding 50%[15–21]. However, these studies also underlined the unreliability of these estimates in the case of TCs, due to the inadequate representation of TC winds and waves in global hindcasts or reanalyses. A few case studies attempted to gauge TC-induced wave setup using high-resolution coastal models, reporting contributions of waves to coastal water elevation ranging from 20 to 100%, depending on local wind and wave forcing, or coastal morphology[22,23]. Yet, a comprehensive, realistic global assessment of the incident wave conditions associated with TCs is still lacking. Addressing this gap is the central objective of the present study.

Furthermore, there is a pressing need to quantify the relative contribution of TCs compared with other phenomena that may generate extreme waves propagating towards coastlines. For example, in

[1]Univ Brest, Ifremer, CNRS, IRD, LOPS, F-29280 Plouzané, France. [2]ENTROPIE (IRD, CNRS, Ifremer, Université de la Réunion, Université de la Nouvelle-Calédonie), Nouméa, New Caledonia. [3]Université de Toulouse, LEGOS (IRD/CNES/CNRS/UT3), Toulouse, France. [4]MARBEC, University of Montpellier, CNRS, Ifremer, IRD, Sète, France. ✉e-mail: swen.jullien@ifremer.fr

the Western Pacific islands, instances of flooding or high sea levels were linked to remote extratropical swells[24,25], while other were associated to local[23] or distant TCs[26]. In the tropical Atlantic, both Southern and remote TC-induced swells were suggested to impact sea level anomalies along the Senegalese coast during boreal summer[27]. Additionally, climate modes of variability, such as El Niño/Southern Oscillation (ENSO), the North Atlantic Oscillation (NAO) or the Southern Annular Mode (SAM), strongly modulate TC and extra-tropical storm distributions[28–30], impacting wave climate variability in the mid-latitudes[29,31] and TC-induced wave height distributions in the Pacific[32–34]. Evaluating coastal hazards thus requires accounting for locally and remotely generated wave systems, along with their inter-annual and seasonal variability.

To address these challenges, we present an innovative study based on twin ocean wave simulations, one incorporating realistic TC forcing and the other without, aiming to achieve two key objectives: (1) assessing the contribution of TCs compared to the contribution of other remotely generated swells to extreme incident waves along global coastlines, including an examination of their interannual variability, and thereby (2) providing reliable extreme offshore wave conditions, essential for evaluating wave-induced setup along global coastlines.

## Results

### Contribution of TCs to extreme waves approaching coastlines

The spatial distribution of extreme wave heights (Hs) along coastlines is illustrated in Fig. 1a with the 98th percentile of Hs at the closest deep-water location for each coastal point, derived from the WAVEWATCH III model's "CYCL" simulation. The highest waves are primarily found in the extra-tropical regions. However, in the tropics, notably in TC-prone regions like the tropical North Pacific, where TC densities are the highest, extreme waves of 4–6 m are also encountered. The "CYCL" simulation, forced with a wind field including observed TC wind amplitude (see Methods for details), closely captures these extreme wave conditions, as indicated by the high correlation coefficient of 0.91 and small mean bias of −0.20 m (-10% underestimation) relative to altimetry observations (Fig. 1b, blue curve, see also Supplementary Fig. S1). When compared to a twin simulation that excludes TC forcing, referred to as "NOCYCL" (see Methods), the simulation accounting for realistic TC forcing is shown to decrease the extreme Hs bias by ~4%, at global scale. Focusing on locations affected by TCs (identified as locations where CYCL and NOCYCL extreme Hs differs) reveals that TC-induced extreme waves are remarkably well represented by CYCL (Fig. 1b, red curve), while the simulation without TCs strongly underestimates extreme waves in these areas.

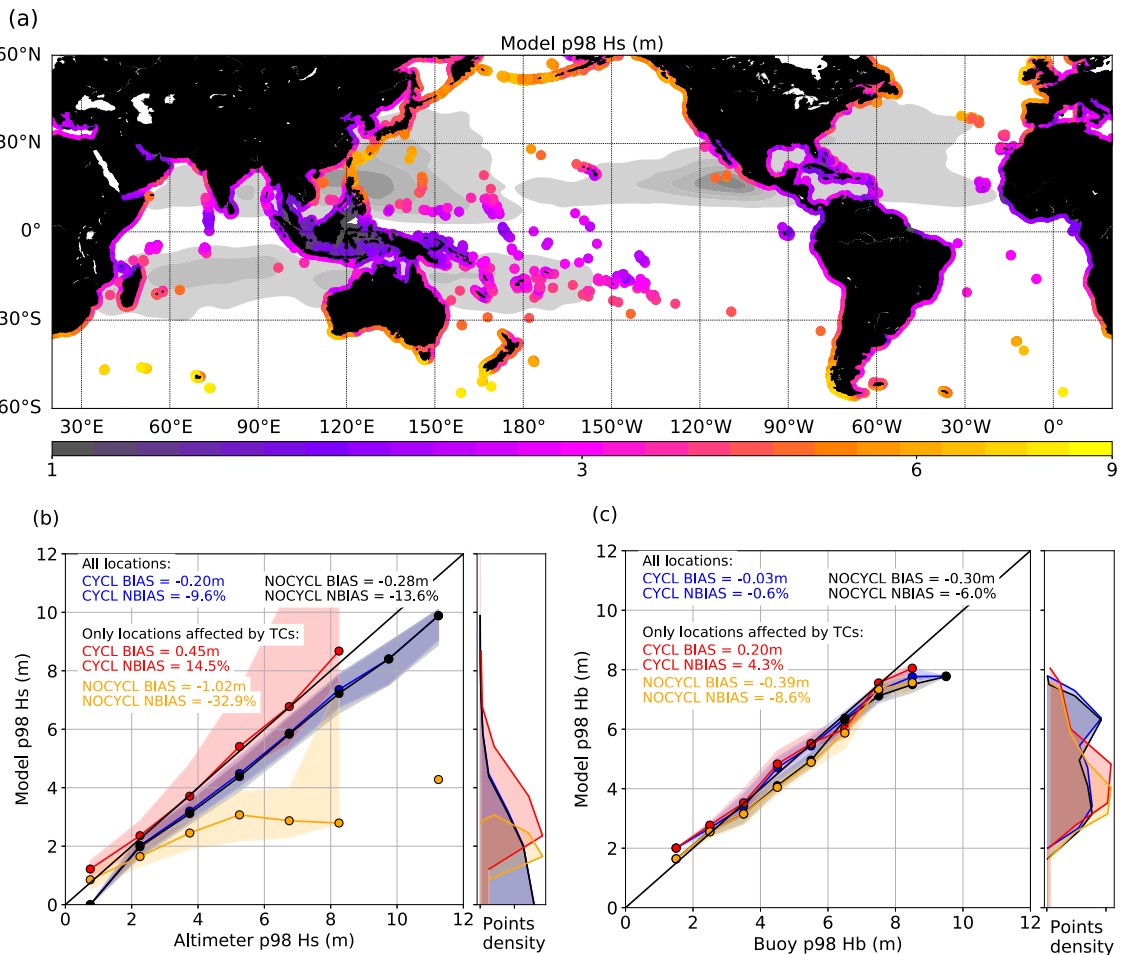

**Fig. 1 | Validation of modeled extreme waves. a** Map of the 98th percentile of significant wave height, Hs (m), at extracted coastal locations (colored dots along the coasts) in the simulation forced with a wind field including observed Tropical Cyclone (TC) wind amplitude (CYCL); **b** simulated versus altimeter 98th percentile of Hs at the extracted coastal locations; **c** simulated versus buoy 98th percentile of breaking wave height, Hb (m), at available buoy locations. In **b** and **c** dots represent the binned median and shading represents the lower-upper quartile interval; all locations are considered for the blue (CYCL, simulation that includes TC forcing) and black (NOCYCL, simulation that excludes TC forcing) curves, while only locations affected by TCs (i.e. when Hs or Hb in CYCL is at least 0.1 m greater than in NOCYCL) are considered for the red (CYCL) and orange (NOCYCL) curves; the density of points is indicated in the side plot.

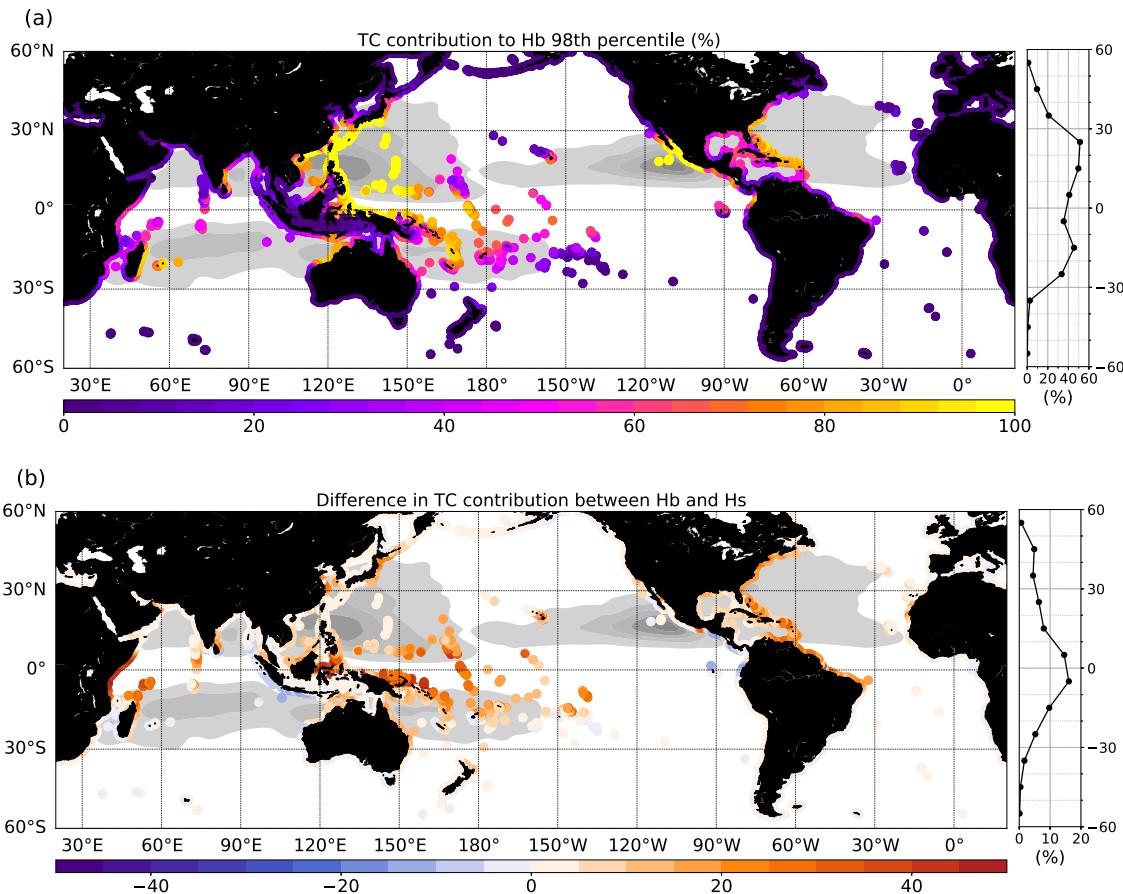

**Fig. 2 | Contribution of Tropical Cyclones (TCs) to extreme wave events. a** The contribution is computed as the relative difference (in %) between the simulation including TC forcing (CYCL) and the simulation excluding TC forcing (NOCYCL) in the number of occurrences above the 98th percentile of breaking wave heights (Hb) over 1990–2017; in **b** the difference in TC contribution when considering Hb (representing the wave energy flux) or significant wave height (Hs) 98th percentile is illustrated. Side plots on the right represent zonal averages of the corresponding map on the left. The cyclone density is indicated with gray shading.

As waves propagate and approach the coasts, their energy flux rather than energy is conserved[35]. To assess wave coastal incident conditions, we therefore use the wave breaking height (Hb), a measure of the wave energy flux (see Methods). A comparison with available in-situ buoys in the North-East Pacific and North Atlantic indicates that Hb is also very well represented by the model (Fig. 1c), with a very weak bias of −0.6% overall and a slightly positive bias of 4.3% in locations affected by TCs. Larger biases arise in the NOCYCL simulation (−6.0% and −8.6%) for respectively all and TC-affected locations. The less pronounced difference between CYCL and NOCYCL than that observed for Hs (Fig. 1b) stems from the fewer number of buoy locations representing the TC-affected areas, in contrast to the numerous coastal locations used for comparison with altimetry (Fig. 1b, c, see also Supplementary Fig. S1).

The TC contribution to extreme wave events approaching the coasts (number of wave occurrences above Hb 98th percentile, Fig. 2a) is then evaluated by comparing the simulations with and without TC forcing. This contribution is shown to be the strongest in the tropical North-Western and North-Eastern Pacific regions, where TCs are particularly frequent and intense. In the core of these regions, incident extreme waves are entirely attributed to TCs. TCs further contribute up to 40–75% in other regions prone to strong cyclonic activity (North-Western Atlantic, China Sea, South Indian Ocean, Bay of Bengal, North-West Australian Coast, South-West Pacific, Hawaii), and to 10–25% in areas where TCs are relatively infrequent (French Polynesia, Arabian Sea). On average, TCs account for 43% of the extreme breaking height events in the tropical band. This contribution is naturally sensitive to

the percentile chosen to define extreme waves: it ranges from 30% for a 95th percentile threshold to 64% when considering a 99.5th percentile threshold (see also Supplementary Fig. S2), emphasizing the growing role of TCs in wave hazards as higher extremes are considered.

### TC remote impacts

It is worth noting that the impact of TCs on wave extremes is not limited to TC-prone regions. Notably, in the equatorial area (10°S-10°N), where TCs are mainly absent, 30% of wave extremes are associated with TCs on average (Fig. 2a). That contribution locally varies from 10 to 90%, being most pronounced along the coasts of the Western equatorial Pacific islands, and the coasts of Papua New Guinea, Indonesia, Ecuador, and Colombia. The East coasts of Africa, stretching from Mozambique to Somalia, are also remotely affected by TCs as well as the Northern coast of Brazil, and of the Cap Verde islands albeit to a lesser extent. This highlights the far-reaching and potentially destructive influence of TCs, even in remote areas.

TC wave-induced hazards as measured by Hb, reflecting the total wave energy flux, significantly differ from hazards solely measured by Hs (Fig. 2b). Indeed, in the TC-prone regions, there is an increase of 5–10% of TC contribution when considering the energy flux rather than Hs. In the inter-tropical region, this difference can be as high as 40% (Fig. 2b). This finding carries strong implications for assessing wave-induced coastal hazards. While wave energy flux is the appropriate metric, Hs is the most globally accessible and validated wave parameter, and represents the most rigorously constrained variable in wave models. On the other hand, Tp measurements are available more

**Table 1 | Global averaged Pearson's correlations of the yearly number of extreme wave events with the El Nino-Southern Oscillation (ENSO) Nino3.4 index, Southern Annular Mode (SAM) index, and North Atlantic Oscillation (NAO) index in the CYCL simulation, in the NOCYCL simulation, and those only associated to Tropical Cyclones (TCs; CYCL-NOCYCL)**

| Correlations of the number of extreme wave events with climate mode indices | | | |
|---|---|---|---|
| | ENSO | SAM | NAO |
| TCs | **0.57 (0.002)** | −0.15 (0.47) | 0.19 (0.35) |
| CYCL | **0.53 (0.006)** | 0.21 (0.29) | 0.03 (0.89) |
| NOCYCL | 0.11 (0.59) | **0.43 (0.03)** | −0.16 (0.45) |

*P*-values are indicated into brackets. Significant correlations are in bold.

sparsely, only for some buoys or from satellite instruments with a lower temporal resolution.

### Interannual variability of TC and non-TC wave hazards

Finally, we investigate the interannual variability of extreme coastal wave events. Table 1 provides the global averaged correlations between key climate modes of interannual variability, and yearly numbers of extreme wave events categorized into TC and non-TC origins. Extreme wave events exhibit a significant correlation with ENSO (0.53, *p*-value = 0.006), especially those associated with TCs (significant correlation of 0.57, increasing to 0.65 when considering the ENSO peak period, November to April, only). During El Niño phases, the Central Pacific faces a significantly higher threat from wave extremes (Fig. 3a), with twice as many events compared to La Niña or Neutral phases. Conversely, regions such as the South China Sea, the Indonesian archipelago, the tropical Atlantic basin, or the northern tip of the Pacific experience an increase in extreme wave events during La Niña phases (Fig. 3a). Changes in TC densities associated to ENSO largely drives the spatial pattern of this interannual variability in coastal wave extremes (Fig. 3b). The area of TC occurrence is shifted eastward in the North-Western Pacific, eastward and equatorward in the South-West Pacific in response to the South Pacific Convergence Zone shift during El Niño[28,36], increasing the exposure of the Micronesian, Melanesian, Polynesian, and Central Pacific Archipelagos. These results align with previous regional studies by Lin et al.[32] and Stephens and Ramsay[33]. During La Niña phases, TC densities and associated extreme waves increase along the North West coasts of Australia, in the Mozambique channel, in the South China Sea, and in the Caribbean region. Wave extremes of non-TC origin are although also modulated by ENSO (Fig. 3c), with an increase of Southern swells propagating across the Pacific basin observed during El Niño phases[37], and an increased number of extreme waves in the Northern Pacific, Eastern Atlantic, and along the Indonesian coasts during La Niña.

While the other modes of interannual variability show globally lower and non-significant correlations with TC wave extremes, it can be noted that the Southern Annular Mode (SAM) exhibits a significant correlation coefficient of 0.43 (*p*-value = 0.03) with extreme wave events of non-TC origin (Table 1). This correlation is particularly significant in the Central Pacific (Fig. 4b), where Southern ocean swells have been shown to significantly contribute to the wave climate[38] and its extremes[24,25]. A negative correlation with TC-induced waves in the low latitudes of the Western Pacific is also observed (Fig. 4a), related to a lower density of TCs in the North-Western Pacific during SAM positive phases. These results are consistent with those of Marshall et al.[39]. Finally, the North Atlantic Oscillation (NAO), which does not hold significant correlation with wave extremes on a global scale, locally exhibits significant impacts. NAO features negative correlations with extreme events in the Caribbean and Southern Europe regions, for both TC and non-TC related events, and a positive correlation at high

latitudes (Fig. 4a, b). These results are in agreement with the regional studies by McCloskey et al.[30] for TCs and Dodet et al.[31] for extra-tropical storms. A positive correlation between TC-induced wave extremes and NAO index is also observed in the North-Western tropical Pacific, in agreement with Choi and Cha[40], who found a positive correlation of TC frequency with NAO in that region, associated with weaker mid and low-tropospheric stream flows and vertical wind shear in the TC-prone area favouring TC genesis during positive NAO phases.

## Discussion

Assessing coastal hazards associated with TCs and extra-tropical storms is vital for effective risk management and preparedness. However, this task presents significant challenges due to a multitude of influencing factors, encompassing wind forcing characteristics, oceanic and wave background conditions, geomorphological factors[3,41], and human aspects[11]. Our study draws a quantitative picture of wave extremes offshore global coastlines, evaluating the respective contributions of TC and non-TC events, along with their interannual variability. Transitioning from offshore wave characteristics to storm inundation on land involves assessing wave transformation at the coast and in shallow water environments, considering factors such as coastline types and morphologies, beach slopes, wave-current interactions, and individual wave motions. Given the heterogeneity of these factors worldwide, such a study cannot be generically conducted on a global scale, and is instead location-dependent. However, while these coastal aspects extend beyond the scope of our study, our findings offer key forcing conditions for coastal dynamical models or statistical parameterizations addressing this issue.

An important highlight from our study is that relying solely on Hs fails to fully capture the incoming wave extremes, especially when dealing with remotely generated swells from both TC and non-TC events. We demonstrate that considering the full energy flux, derived from both the significant wave height and the peak period, enhances the TC contribution to extreme wave events by 5–50% compared to considering Hs alone. Regions untouched by TCs, such as the equatorial band, are consequently found to experience significant impacts from remotely generated extreme waves originating from TCs. This bears important implications regarding the development of future data acquisition strategies. Currently, the most commonly available wave measurements stem from satellite altimetry that provides Hs data. Our results underscore the need for spectral wave measurements to more accurately assess the threats posed by wave extremes. Those are increasingly accessible through recent satellite sensors such as CFOSAT-SWIM[42,43], but remain with a low revisit period (13 days). In-situ direct observing systems also need to be extended, being currently limited to buoys mostly distributed along the coasts of North America and almost absent in the tropics elsewhere (see Supplementary Fig. S1b).

Our study also stresses out the sensitivity of wave extremes to the representation of TCs. Reanalyses such as ERA5 strongly underestimate TC winds (see Supplementary Fig. S3). This leads to a weaker but persistent underestimation of TC-induced waves approaching the coasts. Alternatively, using a wind forcing built by blending reanalysis winds with a parametric TC formulation scaled on TC best-track observations provides extreme waves along coastlines in good agreement with observations. Some limitations remain to be noted. The slight overestimations for waves exceeding 7 m (Fig. 1b) may be attributed to limitations inherent to the TC wind parametric formulation, such as the reconstruction of the radius of maximum winds as a function of latitude and maximum wind[44], as direct estimates are lacking at global scale and over the long period of our study, or the inability to account for fine-scale asymmetries and heterogeneities observed in actual TCs[45]. The horizontal resolution of our wave model (½°) may also smooth and overestimate TC sizes, particularly for the strongest TCs, which have typically smaller sizes. This could result in

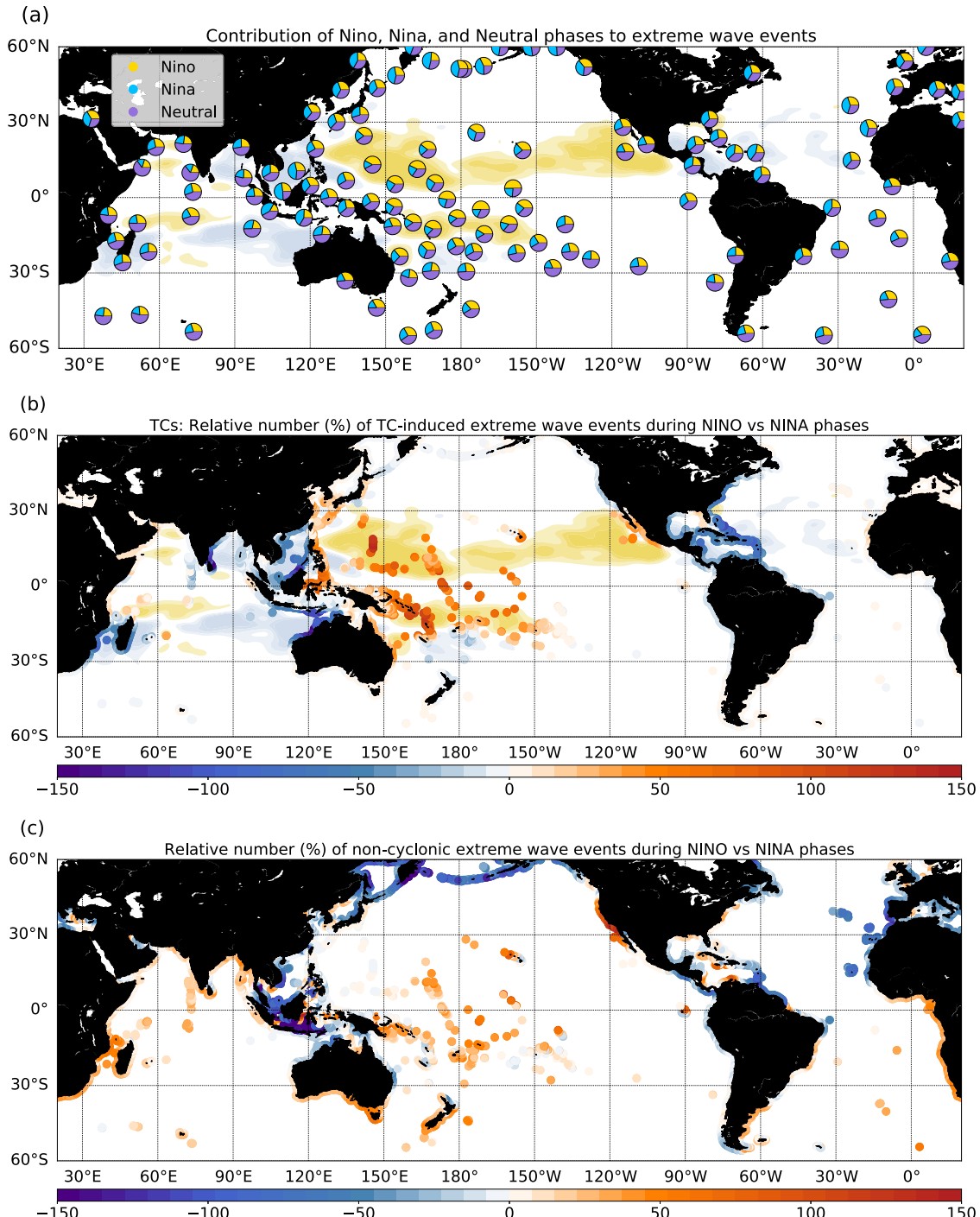

**Fig. 3 | Contribution of El Niño-Southern oscillation (ENSO) phases to extreme wave events. a** Relative fraction of extreme breaking wave height (Hb) events (>98th percentile) associated to Niño (yellow), Niña (blue), or Neutral (violet) phases; **b** increased probability (%) of Tropical Cyclone (TC)-induced Hb extremes during Niño vs. Niña phases; the cyclone density difference between Niño and Niña phases is shaded in gold/lightblue colors; **c** increased probability (%) of non-cyclonic Hb extremes during Niño vs. Niña phases. In **a**, pie-charts are averaged every 200 points for each coastal polygon (see Methods for details on coastal points extraction).

overestimated fetches and an excess of energy available for wave growth[46]. Despite these challenges, our approach provides valuable insights into TC-induced wave extremes, laying the foundation for future research to refine modeling techniques and to enhance our understanding of TC-related hazards under climate change scenarios. Studies attempting to assess projected changes in wave extremes in the future[47] faced the issue of the poor representation of TCs in global climate models. They found diverging results regarding the relative importance of TC or non-TC wave changes depending on the model ability to represent TCs. According to Shimura et al.[47], models with better TC skill suggested twice larger changes for TC than for non-TC waves, the largest change in TC-wave extremes being associated to large changes in TC frequency and intensity. These future TC characteristics however strongly depend on the synoptic conditions favorable to TC genesis and intensification (see for instance Dutheil et al.[48]), which suffer from strong biases in state-of-the art climate

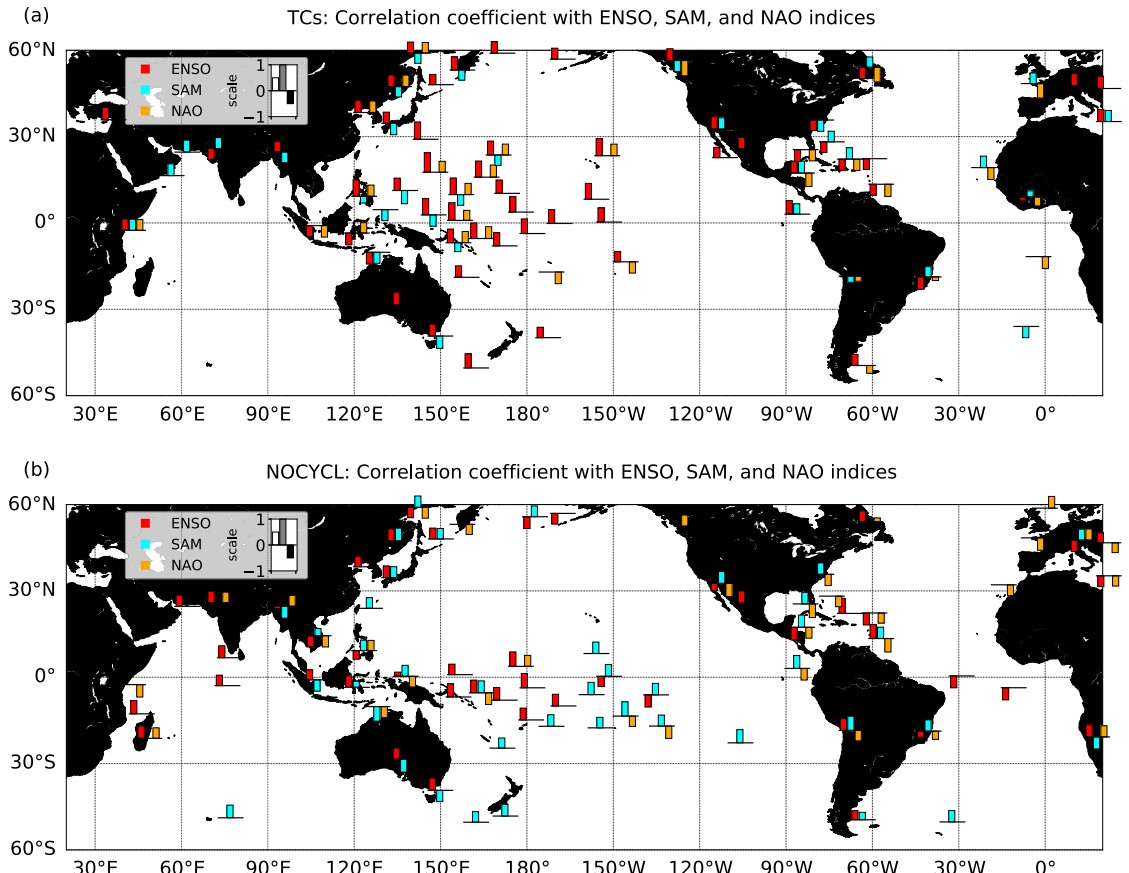

**Fig. 4 | Correlations between climate modes of variability and extreme wave events.** Correlations, when significant above 95%, of the yearly number of extreme events **a** associated to Tropical Cyclones (TCs), **b** associated to non-TC events, with the El Niño-Southern Oscillation (ENSO) Niño3.4 index (red), Southern Annual Mode (SAM) index (cyan), and North Atlantic Oscillation (NAO) index (orange). Bars are displayed averaged every 200 points for each coastal polygon (see Methods for details on coastal points extraction), and only if correlations exceed 95% significance.

models. Given these limitations there is still ample work to be performed to gain confidence in the future extreme wave climate under different future scenarios. Bias correction methods may help improving the representation of TC distributions[48], along with increased resolution. Statistical studies based on future large-scale conditions[49] and wind blending techniques may also help to effectively anticipate and mitigate the impacts of TC-induced hazards on coastal communities worldwide.

## Methods
### Model
The global wave simulations are conducted with the WAVEWATCH III spectral wave model version 6.07 (https://github.com/NOAA-EMC/WW3/releases/tag/6.07.1[50]) on a 0.5° rectangular grid (limited to 78°S and 80°N). The compulsion of using a relatively coarse grid resolution arose from the limitation of high computational cost of the model running globally for a long duration (1990–2017). The spectral space is discretized in 24 directions (15° resolution), and 32 frequencies ranging from 0.0373 to 0.7159 Hz with an increment factor of 1.1. The frequency range allows to solve the main swell peaks and wind-induced waves. A diagnostic tail following a $f^{-5}$ power law[51] is added at higher frequencies. Beyond the lower frequency, we use the parameterization of Ardhuin et al. [52], which has been shown to effectively represent the characteristics of free infra-gravity (IG) waves, particularly relevant for highly energetic events like those examined in our study. The ultimate quickest third order propagation scheme is used for both spatial and spectral advection, with the garden sprinkler effect correction proposed by Tolman[53], limiting the impact of the

directional discretization. Non-linear wave-wave interactions are modeled using the discrete interaction approximation of Hasselman et al. [54]. The bathymetry used is GEBCO 30" in its 2014 version (http://www.gebco.net/data_and_products/gridded_bathymetry_data/). Sub-grid scale islands are addressed through an obstruction parameterization, which effectively reduces the wave energy of the cell in proportion to the obstruction[55,56]. Reflection at the coast is accounted for with a coefficient of 0.1 and with a dependence to frequency[57]. SHOWEX bottom friction[58] and depth-induced breaking parameterization[59] with a Miche-style shallow water limiter for maximum energy are used. For wave growth and wave dissipation the ST4 source term package[60] is chosen, with the wind-wave growth parameter, $\beta_{max}$, adjusted to 1.6, the sheltering coefficient set to 0.3, and the swell dissipation parameters set to SWELLF = 0.69, SWELLF3 = 0.022, SWELLF4 = 150000, SWELLF7 = 468000. The wind forcing is detailed in the following section. We do not use any wave-ice interactions as the focus of the study is on the tropical and extra-tropical regions, but we use a daily evolving ice mask from the Climate Forecast System Reanalysis (CFSR[61],). We focus on the 60°S-60°N region for our analyses. WAVEWATCH III uses a fractional time step method with 4 time steps. The global time step treating temporal variations of the depth is set to 2400 s, the time step for spatial propagation is 480 s, the time step for refraction is 300 s, and the minimum time step for integrating the source terms is 10 s.

### Experiments
The forcing wind fields are based on the European Centre for Medium-Range Weather Forecasts fifth generation atmospheric reanalysis of

the global climate (ERA5, DOI: 10.24381/cds.adbb2d47). ERA5 is distributed on a 0.25° atmospheric grid and with hourly frequency. This reanalysis features a reduced negative bias in wind speed compared to previous reanalyses[62], and was shown to improve TC storm surge simulations. However, a significant bias still remains for intense, small size, and fast moving TCs. The overall bias in cyclone maximum wind speeds provided in the ERA5 reanalysis is −37% compared to the Best Tracks observations (see Supplementary Fig. S3). To better represent the effects of TCs, we replaced the ERA5 surface wind in TCs by a parametric structure reconstructed from the observed maximum wind speed following a procedure similar to that of Vincent et al. [63]. We first filtered out the weaker than observed TCs in ERA5 with an 11-day time running-mean filter within 600-km of each TC track position. Then a 2D surface parametric wind field, reconstructed using the observed maximum wind speed in TCs provided by K. Emanuel Best Tracks database (ftp://texmex.mit.edu/pub/emanuel/HURR/tracks/) and the parametric model of Willoughby et al.[44], was added to the filtered wind field at each cyclone position between 0 and 600 km from the track. The parametric model was calculated so that the amplitude reached at the TC center in the reconstructed forcing was equal to the observed intensity, and using a radius of maximum TC winds ideally derived as a function of latitude and maximum wind[44]. To avoid a rough transition at 600 km between the new wind field including the TC winds (referred to as wind$_{new}$ in the following) and the ERA5 native wind field, a linear transition was applied between 600 and 1200 km, such as $wind = (1-\alpha)$ $wind_{new} + \alpha.wind_{ERA5}$, with $\alpha$ being a linear function ranging from 0 at 600 km from the TC track to 1 at 1200 km. ERA5 wind field remains untouched off 1200 km of the TC tracks. Such procedure has been shown to be very successful in reproducing the observed ocean hydrodynamics under TCs[63].

Two WAVEWATCH III simulations are performed over the 1990–2017 period with two different forcing fields: the ERA5 surface winds merged with parametric TCs of observed amplitudes (referred to as "CYCL"), and the ERA5 surface winds with filtered TCs (referred to as "NOCYCL"). The difference between CYCL and NOCYCL simulations is used to isolate the impact of TCs on coastal extreme sea states, while NOCYCL simulation is used to assess the impact of non-TC events.

## Extraction of coastal locations

As the shoreline morphology is poorly known in numerous locations and additionally may evolve in time (erosion, deposition), the assessment of the contribution of waves to the water level at a global scale is very difficult (e.g. see discussions of refs. 19,64–67). We therefore only consider deep-water incident wave conditions, hence avoiding any other hypothesis.

The simulation results are depicted along global coastlines and islands as defined by Smith and Sandwell[68], albeit sub-sampled to enhance visual clarity and accommodate the relatively coarse resolution of our model. Sub-sampling is performed as follows: the coastline data[68] are first re-gridded on a 1/24° grid, followed by sub-sampling of the coastline based on the size of each coastal polygon. For large polygons (more than 60 points), every 6 points (~25 km) are retained, while for medium-sized polygons (between 15 and 60 points), every 3 points (~13 km) are retained. All locations (~4 km) are preserved for small polygons (less than 15 points). Subsequently, the resulting coastline undergoes meticulous visual inspection to eliminate any undesired closed contours that may have appear in rugged coastline areas. Finally, wave fields are extracted at the nearest deep-water location for each of these coastal points. Typical offshore distance along major continents is about 50 km according to our ½° model resolution, with a few points showing larger distance due to specific rugged coastlines not properly accounted for in our relatively coarse resolution model mask.

## Wave energy flux and parameters of interest

The main parameters used are the significant wave height (Hs), and the peak period (Tp). Hs represents the integral of the wave spectrum energy E: $Hs = 4\sqrt{E}$.

The wave energy flux (WEF), quantifying the amount of energy carried by ocean waves as they propagate, is computed as:

$$WEF = EC_g \tag{1}$$

with $E = \frac{1}{2}\rho g a^2$ the wave energy, $a$ being the wave amplitude, $\rho = 1025$ kg.m$^{-3}$ the water density, $g = 9.81$ m.s$^{-2}$ the gravitational acceleration, and $C_g$ the wave group velocity. When waves propagate and approach the shore, the flux, rather than the energy, is conserved, making WEF a more relevant variable than Hs to assess wave coastal incident conditions. For an easier comparison of Hs and WEF at our coastal points, we use the breaking wave height, Hb, as a measure of the wave energy flux (WEF). It is computed assuming that wave breaks at a depth equal to the wave height[69], and considering that WEF is conserved from deep water to the depth of breaking:

$$WEF_{offshore} = WEF_b \tag{2}$$

with WEF$_{offshore}$ the WEF of deep water offshore condition, computed with $C_g = \frac{g}{4\pi}T_p$ the group velocity in deep-water, and $a = \frac{Hs}{2\sqrt{2}}$ the wave amplitude in deep-water:

$$WEF_{offshore} = EC_g = \frac{\rho g^2}{64\pi}H_s^2 T_p \tag{3}$$

and WEF$_b$ the WEF at the depth of breaking, computed with $C_g = \sqrt{gH_b}$ the group velocity in shallow-water, and $a = \frac{H_b}{2\sqrt{2}}$ the wave amplitude at the breaking depth:

$$WEF_b = EC_g = \frac{\rho g^{3/2}}{16}H_b^{5/2} \tag{4}$$

Therefore, Hb writes:

$$H_b = \left(\frac{H_s^2 \sqrt{g}}{4\pi}T_p\right)^{\frac{2}{5}} \tag{5}$$

## Statistical estimations and contributions

The 98$^{th}$ percentile of a field F is used as a threshold for extreme events, and referred hereafter to as F$_{REF}$. The choice of this threshold is discussed in the results. This threshold is computed in the CYCL simulation. Then, the occurrence of extremes in a simulation is computed as the number of time steps N where F reaches or exceeds F$_{REF}$.

The contribution of TCs to extreme events is then evaluated by computing the relative difference in the number of occurrences above the threshold in the CYCL (N$_{CYCL}$) and NOCYCL (N$_{NOCYCL}$) simulations, and is given in percentage:

$$TC_{contrib} = (N_{CYCL} - N_{NOCYCL})/N_{CYCL} * 100 \tag{6}$$

## Indices of climate mode of variability

The correlation between the TC contribution to extreme waves and various modes of climate variability is computed. The North Atlantic Oscillation (NAO) index is extracted over the period of the simulations from https://www.cpc.ncep.noaa.gov/products/precip/Cwlink/pna/nao.shtml, the Southern Annular Mode (SAM) index from https://www.cpc.ncep.noaa.gov/products/precip/Cwlink/daily_ao_index/aao/

aao.shtml, and the Nino3.4 index from https://www.esrl.noaa.gov/psd/data/climateindices/list/. Yearly values of the indices and of the TC contribution are used to compute the correlations. The relative TC contribution to wave extremes during Niño vs. Niña years is also evaluated. To do so, the monthly values of the Nino3.4 index are considered. ENSO phases are computed as Niño (resp. Niña) phases defined for Nino3.4 > 0.4 (resp. Nino3.4 < −0.4) and neutral phases for Nino3.4 in [−0.4, 0.4]. Each simulation time-step is then attributed to an ENSO phase, and the number of events are summed for each phase. For comparing TC contribution to extreme waves during El Niño and La Niña phases, the number of time-steps are weighted by the total number of respective Niño and Niña phases.

### TC densities
A probability density function (PDF) is used to illustrate the spatial distribution of TC occurrence. The PDF is computed as the normalized sum of anisotropic Gaussian functions for each TC occurrence, with standard deviations in the meridional and zonal directions respectively of 1.5° and 3°, as in refs. [70],[71].

The cyclone density difference between El Niño and La Niña phases is computed as the difference between TC densities summed over Niño and Niña phases, and weighted by the total number of respective phases.

### Observed Hs and Hb
The sea state Climate Change Initiative satellite dataset[72] is used for validation of the model significant wave height (Hs). This dataset provides along-track inter-calibrated and denoised[73] estimations of Hs from all available altimeter measurements over the 1992–2017 period. Coastal values in a 50 km along-shore area are flagged out because of the poor reliability of the data due to land interference with the signal. Inaccurately measured values are also flagged out based on the noise of the retrieved signals. In order to perform a proper comparison, both model outputs and along-track satellite Hs are extracted in a 0.5° × 0.5° box on our coastal-extracted points (which are ~50 km offshore). Model points are considered only when a satellite measurement is available. The annual mean and 98th percentile of Hs are computed for each coastal point and each simulated year, as well as their correlation, bias, normalized bias, normalized root mean square error, and scatter index.

The Copernicus Marine Service (CMEMS) In Situ Thematic Assembly Center (INSTAC) buoy data (http://www.marineinsitu.eu/) are used for validation of the model breaking height (Hb). Among all the in situ wave buoys available, not all of them record both the Hs and Tp parameters, required to compute Hb. We therefore uses a subset of buoys which were recording both variables during at least one year over the period 2012–2016. We also keep only buoys located at least 50 km offshore, and moored in more than 50 m water depth to be consistent with our modeled data. 70–80 buoys were fitting these criteria for the 5-year period of validation.

### Data availability
All data used in this study are available online or from the corresponding author upon request. Simulation wave fields extracted at the nearest deep-water location for each of the coastal points are available at: https://doi.org/10.12770/25312128-f0b5-46ca-9f14-3bb38469ff05 along with the processed data used in the Figures. Observational data are available from the different providers: for K. Emanuel TC Best Tracks data at ftp://texmex.mit.edu/pub/emanuel/HURR/tracks/, for the CCI satellite dataset at: ftp://anon-ftp.ceda.ac.uk/neodc/esacci/sea_state/data/v1.1_release/, for the CMEMS INSTAC buoy data at http://www.marineinsitu.eu/. The climate indices are available: for NAO at https://www.cpc.ncep.noaa.gov/products/precip/Cwlink/pna/nao.shtml, for SAM at https://www.cpc.ncep.noaa.gov/products/precip/

Cwlink/daily_ao_index/aao/aao.shtml, and for Nino3.4 at https://www.esrl.noaa.gov/psd/data/climateindices/list/.

### Code availability
The simulations were run with the WAVEWATCH III spectral wave model version 6.07.1[50], publicly available at https://github.com/NOAA-EMC/WW3/releases/tag/6.07.1. Parameterizations and settings used in the study are described in the Method section. The analyses and figures were performed with Matlab and Python scripts, available from the corresponding author upon request.

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

## Acknowledgements

The simulations were performed by S.J. using HPC resources from GENCI-IDRIS (Grant A0010107661) and the Pôle de Calcul et de Données Marines (PCDM; http://www.ifremer.fr/pcdm). The study was supported by the French National Institute for Ocean Science (Ifremer) for S.J., the French Nation Research Institute for Sustainable Development (IRD) for C.M., M.L., J.A., E.K., the LEFE program (CTroVagueS project obtained by S.J.), the French National Space Agency (CNES, LAGOON project focused on CFOSAT mission, obtained by S.J.), and the French National Research Agency (grant ANR-22-POCE-0002).

## Author contributions

S.J., C.M. and J.A. designed the study. S.J. performed the simulations. S.J. and E.K. extracted the data, and coded the analyses. S.J., C.M., J.A., M.L., E.K. contributed to interpreting and discussing the results, drafting and revising the manuscript.

## Competing interests

The authors declare no competing interests.
