## [Peer Review File · Nature Communications]

Unveiling the Global Influence of Tropical Cyclones on Extreme Waves approaching Coastal AreasREVIEWER COMMENTS

Reviewer #1 (Remarks to the Author):

This study investigated the tropical cyclone (TC) contribution on ocean wave energy flux along global coast in comparison between long-term wave hindcast with and without tropical cyclone. This study firstly showed that the significance of TC contribution in global scale. The contribution has been overlooked in the context of global ocean wave climate study. Therefore, the quantitative estimation of TC contribution by this study has impact on wide variety of fields related coastal process, such as estimation of climate change impact assessment on coastal hazard. I recommend this study be published after revision. The detail comments are below.

--- Major comment

My main concern is the lack of validation of wave period and wave energy flux. The validation on significant wave height has been done and the performance looks good. The highlight of this study is the TC contribution on energy flux not significant wave height. The parametric TC model has been well validated for significant wave height around TC in the previous studies but the validity of swell field propagating to far field has not been well studied. So, I recommend the authors to validate the period and energy flux of waves propagating to far-field using buoy observations. No need to pick up all the locations, but I think the observation by NDBC buoys at Hawaii and east coast of US are good candidate locations because Figure 2 shows the relatively large TC contribution.

--- Minor comments

Figure 1c should be the same style as Figure 1b. Why you change the style?

Line 101-104. Putting the figures for 95th percentile and 99.5th percentile in the Supplement materials are helpful.

Same as Figure3b but a panel for nonTC should be added to show the ENSO variability is dominated by TC wave variability.

Figure 3a and 4. How did you select the location for the plot?

Line 146. SAM seems to have significant effect on TC waves in western low-latitudes Pacific. Please discuss this mechanism.

Line 291. Dp is used?

Reviewer #2 (Remarks to the Author):

“Unveiling the global influence of tropical cyclone extreme waves on coastal vulnerability”

The authors have analyzed the impact of tropical cyclones (TC) on extreme wave activity using model simulations on a global scale. This simulations study has enabled the quantitative assessment of TC's role in extreme waves reaching global coastlines. The authors have then quantified the contribution of the TCs to extreme breaking heights in the tropical regions and identified the amplification of TC-induced vulnerability of central Pacific during El Niño and south China Sea, Caribbean arc and the south Indian ocean coastlines during the La Niña events.

The manuscript covers the objectives fairly well, while lacking severely in some aspects. There is a significant need of grammar corrections (e.g. Lines 95, 238) throughout the manuscript to make it more comprehensible. In the current form, the manuscript requires major revisions, and I recommend another round of reviews of the revised version.

Some specific comments are listed below. The manuscript requires all the more attention by the authors given the significance of the targeted journal and its audience.

Section-wide Concerns:

1. The title suggests the study to be an assessment of coastal vulnerability impacted by cyclone-induced extreme waves. However, the manuscript is mainly focused on cyclone-induced deep water wave activity away from coasts, and does not directly infer any implications on the coast. It is suggested to use a more appropriate title.
2. The “Introduction” part does not clearly portray a well-put flow of motivation, rather jumps from one fact to another before stating the objectives. A clear flow of motivation with precise gap areas is needed.
3. The “Discussion” section repeats many points stated in other sections, and reiterates the existing gap in literatures, where it should have instead highlighted novel findings and implications of the current study. It is strongly suggested to rewrite the section keeping in mind the above points.
4. The “Methods” section provides useful insights into various aspects of the techniques used in the study. However, it fails to convey some very critical aspects of model configuration, and choice of algorithms, which makes the analysis seem weakly founded.

Line-specific Concerns:

1. [Line 33] Missing bracket in the line, as corrected here: “... ecosystems, type of sediment [7][3] ...”
2. [Lines 37 – 40] Should be rephrased to make it more comprehensible.
3. [Line 44] Suggested to rephrase as “... water levels, with this contribution ...”.
4. [Line 77] It has not been mentioned whether the closest deep-water locations for each coastal points are beyond 50 kms away from the coast, as required for comparison with the satellite data.
5. [Line 78] Consistent use of either WaveWatch-III or WWIII is recommended.
6. [Line 103] Should be written as “... sensitive to the percentile ...”.
7. [Line 147] Suggested to rewrite as “... it can be noted that SAM exhibits a correlation coefficient of 0.43 with extreme wave ...”.
8. [Line 233] Use of the word “prohibited” is unsuitable in the context. Suggested to rewrite as “The

compulsion of using a coarse resolution arose grid was owing to the limitation of high computational cost of the model running globally for a long duration (1993-2017).”, or in this lines.

9. [Line 234] Does the study period span the entire duration of 1993 – 2017, or only for the specific cyclone durations in this interval? The authors should add a clear statement in the text describing the study period.

10. [Line 234] A justification for the choice of spectral space resolution and frequencies is suggested to be added here to introduce more clarity and foundation to the study.

11. [Line 235] Simulation time-steps should be clearly mentioned somewhere in the text.

12. [Line 239] Use of vague statements, such as “... mostly similar to ...”, are not recommended. Suggested to describe the parameterization schemes briefly.

13. [Line 244] Either proper justification or citation is required to support the statement.

14. [Lines 245 – 247] Suggested to rewrite the sentence.

15. [Line 252] A list of the atmospheric variables used as forcings to the model will make the manuscript more informative.

16. [Lines 254 – 258] The sentence is too long and incoherent. Suggested to use shorter, clear sentences.

17. [Line 262] What is the justification of filtering TCs by using 11-day running-mean technique, when more sophisticated methods of removing TCs and calculating background circulation exist? Have the authors looked into calculating background winds using the stream function and velocity potential [1] or using the weighted mean of winds through the atmosphere between 850 hPa and 250 hPa levels [2]? How different will the inferences of this study be if the authors would use either of these two methods?

[1] Arakane, S., Hsu, HH. A tropical cyclone removal technique based on potential vorticity inversion to better quantify tropical cyclone contribution to the background circulation. *Clim Dyn* 54, 3201–3226 (2020). <https://doi.org/10.1007/s00382-020-05165-x>

[2] Emanuel, K., Ravela, S., Vivant, E., & Risi, C. (2006). A statistical deterministic approach to hurricane risk assessment. *Bulletin of the American Meteorological Society*, 87(3), 299-314. <https://doi.org/10.1175/BAMS-87-3-299>

18. [Line 263] What is the justification of using a linear transition of winds between 600 and 1200 kms away from the cyclone tracks instead of using the actual observations?

19. [Line 284] Is the subsampling of coastal points random, or have the authors followed any specific criteria to subset the coastal points?

20. [Lines 297 – 300] It is suggested that the authors rewrite this part to bring in more clarity.

21. [Lines 303 – 304] On what basis have the authors chosen 98th percentile as the threshold for extreme events?

22. [Line 317] It is known that the parameters behind NAO, SAM and Nino3.4 have intraseasonal to yearly oscillations. What is the justification of comparing yearly averages of these indices with contribution of TCs which are highly seasonal?

23. [Line 321] Is the criteria that Nino3.4 index should be more than 0.4 (less than -0.4) for at least six consecutive months followed in the calculation of ENSO phases? If yes, it may be mentioned in the text.

24. [Line 325] The line may be rewritten as “... number of respective Niño and Niña phases.”

25. [Line 327] The term CCI should be expanded at first occurrence.

26. [Line 331] The line may be rewritten as “Inaccurately measured values ...”.
27. [Line 534] On what basis have the authors chosen the 0.3 m threshold while defining the TC impact?
28. [Line 538] Inconsistencies in the choice of study period (1990 – 2017 or 1993 – 2017) must be resolved.
29. [Line 544] How did the authors calculate the cyclone density difference between Niño and Niña phases?
30. [Line 549] The line should be written as “... events with the Niño3.4 ...”.

Reviewer #3 (Remarks to the Author):

This manuscript studies extreme waves generated by tropical cyclones. They claimed by designing twin oceanic wave simulations with and without realistic TC wind forcing from ERA5 (with filtered 2D parametric wind), they enable to quantify TC’s role on extreme waves for the first time. They found out waves substantially contribute to extreme breaking heights in tropical regions (35-50% on average), reaching 100% in high-density TC areas like the North Pacific. TCs also impact remote TC-free regions, such as the equatorial Pacific experiencing on average 30% of its extreme wave events due to TCs.

However, the claimed novelty of the paper is not clear. The following paper “Seamless projection of global storm surge and ocean waves under a warming climate” modeled waves with WAVEWATCH III (the model used in this study) globally. They also considered the future projection. So it is not new to model waves globally. The other novelty the authors claim is replacing the TC wind fields in reanalysis with parametric wind field. It is well known that reanalysis cannot well resolve TCs, and previous studies (included one cited by the authors) has done replacing TC wind fields in reanalysis as in this research. It is also well known that TCs dominate extreme levels in high-density TC areas. Maybe the finding that tropical regions relatively far away from TC areas also show TC induced waves. However, the authors didn’t explain why. In fact, this analysis focus on waves in deep ocean, 50 km away from the coast; it is not clear if these remote effect of TCs will reach the coastal areas. One thing the authors can do is simple check coastal storm surge and wave records and see if remote (with “remote” to be clearly defined) coastal areas are affected by TCs.

There are also many confusions. The authors keep saying coastal impact and storms surge and wave setup (in motivation and implication) but they only studied the waves in deep ocean, which is far from the coast. In the discussion section, it is mentioned that “Numerous prior studies have highlighted the shortcomings in storm surge assessments, in relation with the poor representation of TC winds and extreme waves in global models [9-11][13- 14][17][22][38].” How is the connection of the wave study that has been done here with storm surge? Storm surge is the rise in sea level along the coastline. How can these waves impact the storm surge? Are they studying deep-water waves, which could differ from the effects of nearshore waves?

The motivation started with tropical and extra-tropical cyclones, but didn't continue to discuss about extra-tropical cyclone waves.

Please explain the physical meaning of wave energy flux and why it is an important metric for what.

Not clear how they modeled the wind field. Historical TC storm track and intensity information are available globally. Why do you need to use Emanuel database for observed maximum wind? Storm size information is only available for Atlantic, so how do you estimate size for other regions (L205 needs explanation and reference. L213, "overestimate TC size" by which model?). Given the storm parameters, how do you model the wind field?

It is well known that climate state represented by index like ENSO affects TCs and thus TC induced waves. What is the implication of the results in this component?

There are numerous language errors. Some examples:

L38, "wave setup, i.e., ..., is often lacking.

L54-57, quoting a very long and confusing sentence from others

L70-73, Objectives of the study are very confusing (and about coastal and coastlines). "assessing the contribution of TCs compared to remotely generated swells..." (contribution of TCs compared to contributions of remotely generated swells???) "Providing reliable extreme offshore wave conditions which will serve as a valuable resources for...(how do you provide wave conditions?)

L116-121. Note "vulnerability" is related to the system or coastline affected by the waves, it's the inability of the system to withstand the hazards. "wave-induced vulnerability" is very confusing.

L245-246 "[56]parametrization..., ..., ..., is..."

L267. Please explain the linear transition.

L278. "As the knowledge of ...and evolves in time,..." (knowledge evolves in time???)

Shimura, T., Pringle, W.J., Mori, N., Miyashita, T. and Yoshida, K., 2022. Seamless projections of global storm surge and ocean waves under a warming climate. *Geophysical Research Letters*, 49(6), p.e2021GL097427.

Reply to Reviewer #1 comments

Reviewer #1 (Remarks to the Author):

This study investigated the tropical cyclone (TC) contribution on ocean wave energy flux along global coast in comparison between long-term wave hindcast with and without tropical cyclone. This study firstly showed that the significance of TC contribution in global scale. The contribution has been overlooked in the context of global ocean wave climate study. Therefore, the quantitative estimation of TC contribution by this study has impact on wide variety of fields related coastal process, such as estimation of climate change impact assessment on coastal hazard. I recommend this study be published after revision. The detail comments are below.

We thank the reviewer for his/her comments on our manuscript. We have implemented all the suggestions.

--- Major comment

My main concern is the lack of validation of wave period and wave energy flux. The validation on significant wave height has been done and the performance looks good. The highlight of this study is the TC contribution on energy flux not significant wave height. The parametric TC model has been well validated for significant wave height around TC in the previous studies but the validity of swell field propagating to far field has not been well studied. So, I recommend the authors to validate the period and energy flux of waves propagating to far-field using buoy observations. No need to pick up all the locations, but I think the observation by NDBC buoys at Hawaii and east coast of US are good candidate locations because Figure 2 shows the relatively large TC contribution.

We performed a validation of the breaking wave height (H_b), which serves as proxy for the wave energy flux, using data from available buoys, primarily sourced from NDBC buoys. These buoys were carefully selected to be positioned at least 50km offshore and water depths exceeding 50m, aligning with the deep-water criterion of our study. Throughout the period 2012-2016, approximately 60 to 80 buoys measuring H_s and T_p were available each year. Our analysis reveals a very good agreement between the model and buoy data for the 98th percentile of H_b (Fig. R1). CYCL simulation exhibits a normalized bias of 0.6%, a correlation of 0.96 and a normalized root mean square error of 8.7%. When focusing solely on locations affected by TCs, the normalized bias is slightly positive but remains weak in our CYCL simulation (4.3%). Larger biases arise in the NOCYCL simulation (-6.0% and -8.6%) for respectively all and TC-affected locations. The difference between CYCL and NOCYCL in locations affected by TCs appears less pronounced than that observed when comparing H_s owing to the fewer number of buoy locations representing these areas, in contrast to the numerous coastal locations used for comparison with altimetry (see new Fig. 1b,c of the manuscript). The normalized density distributions of H_b at buoy locations also reveal a shift towards higher values when considering all locations compared to focusing solely on TC-affected areas, underscoring the significant contribution of extra-tropical swells.

These results underscore the model's ability to accurately simulate the extreme wave energy fluxes propagating towards coastal zones. Fig R1a was added to the manuscript as Fig. 1c along with its description, and Fig R1b is added to the supplementary material (S1b).

Fig. R1: Comparison between the simulated 98th percentile of Hb in CYCL simulation and buoy data over the 2012-2016 period. (a) Dots represent the binned median with shading indicating the lower-upper quartile interval; the density of points, normalized over each Hb distribution, is indicated in the side plot. Both blue (CYCL) and black (NOCYCL) curves consider all buoys locations, while the red (CYCL) and orange (NOCYCL) curves exclusively consider locations impacted by TCs (i.e. when Hb in CYCL is at least 0.1m greater than in NOCYCL). (b) Map depicting the normalized bias at each buoy location.

--- Minor comments

Figure 1c should be the same style as Figure 1b. Why you change the style?

The style has been standardized for consistency.

Line 101-104. Putting the figures for 95th percentile and 99.5th percentile in the Supplement materials are helpful.

These figures are now included in the supplementary material.

Same as Figure3b but a panel for nonTC should be added to show the ENSO variability is dominated by TC wave variability.

Such a panel is now included in Fig. 3. A couple sentences have also been added in the text to discuss results from this new panel.

Figure 3a and 4. How did you select the location for the plot?

The sub-sampling method for plotting pie-charts or bars involves averaging these pie-charts or bars every 200 coastal points. Furthermore, in Fig. 4, bars are exclusively displayed if the correlations exceed 95% significance. This is now detailed in the figure captions.

This sub-sampling is implemented for clarity purpose. However, all pie-charts or bars are accessible and users can opt for any sub-sampling or zoom as needed. A public link and DOI for simulated wave fields, extracted at the nearest deep-water location for each of the coastal points, used in the study will be provided (data are in the process of being stored in a public repository).

Line 146. SAM seems to have significant effect on TC waves in western low-latitudes Pacific. Please discuss this mechanism.

SAM has indeed a significant negative correlation with TC-induced waves in the low latitudes of the western Pacific. This is associated with a lower density of TCs in the north-western Pacific during SAM positive phases (see Fig. R2). These results align with those of Marshall et al. (2018), who found lower (resp. higher) probabilities of Hs above the 95th percentile during SAM positive (resp. negative) phases in the low latitudes of the western Pacific during the season favorable to TCs (September-October-November, see their Fig. 11a,b). This is now discussed in the manuscript:

“A negative correlation with TC-induced waves in the low latitudes of the Western Pacific is also observed, related to a lower density of TCs in the North-Western Pacific during SAM positive phases. These results are consistent with those of [40].”

Fig. R2: Difference between positive and negative phases of SAM of the PDF of TC occurrence (in number of days per 5°x5° per 20 years).

Line 291. Dp is used?

No, in the paper only Hs and Tp are used. The sentence has been corrected.

Reply to Reviewer #2 comments.

Reviewer #2 (Remarks to the Author):

“Unveiling the global influence of tropical cyclone extreme waves on coastal vulnerability”

The authors have analyzed the impact of tropical cyclones (TC) on extreme wave activity using model simulations on a global scale. This simulations study has enabled the quantitative assessment of TC's role in extreme waves reaching global coastlines. The authors have then quantified the contribution of the TCs to extreme breaking heights in the tropical regions and identified the amplification of TC-induced vulnerability of central Pacific during El Niño and south China Sea, Caribbean arc and the south Indian ocean coastlines during the La Niña events.

The manuscript covers the objectives fairly well, while lacking severely in some aspects. There is a significant need of grammar corrections (e.g. Lines 95, 238) throughout the manuscript to make it more comprehensible. In the current form, the manuscript requires major revisions, and I recommend another round of reviews of the revised version.

We thank the reviewer for his/her valuable feedback on our manuscript. We have incorporated his/her suggestions to enhance clarity.

Some specific comments are listed below. The manuscript requires all the more attention by the authors given the significance of the targeted journal and its audience.

Section-wide Concerns:

1. The title suggests the study to be an assessment of coastal vulnerability impacted by cyclone-induced extreme waves. However, the manuscript is mainly focused on cyclone-induced deep water wave activity away from coasts, and does not directly infer any implications on the coast. It is suggested to use a more appropriate title.

We have changed the title to: “Unveiling the global Influence of tropical cyclones on extreme waves approaching coastal areas”

2. The “Introduction” part does not clearly portray a well-put flow of motivation, rather jumps from one fact to another before stating the objectives. A clear flow of motivation with precise gap areas is needed.

We have revised the introduction to clarify the motivation, the gaps in the literature, and the objectives of the study.

3. The “Discussion” section repeats many points stated in other sections, and reiterates the existing gap in literatures, where it should have instead highlighted novel findings and implications of the current study. It is strongly suggested to rewrite the section keeping in mind the above points.

We have revised the discussion following the reviewer's suggestions.

4. The “Methods” section provides useful insights into various aspects of the techniques used in the study. However, it fails to convey some very critical aspects of model configuration, and choice of algorithms, which makes the analysis seem weakly founded.

We have extended the Method section to provide more details, according to the reviewer's comments and suggestions.

Line-specific Concerns:

1. [Line 33] Missing bracket in the line, as corrected here: "... ecosystems, type of sediment [7][3])..."

Corrected.

2. [Lines 37 – 40] Should be rephrased to make it more comprehensible.

Rephrased.

3. [Line 44] Suggested to rephrase as "... water levels, with this contribution ...".

Corrected.

4. [Line 77] It has not been mentioned whether the closest deep-water locations for each coastal points are beyond 50 kms away from the coast, as required for comparison with the satellite data.

The extracted point are indeed at 50km away from the coast or beyond, as stated in the Methods section:

"Typical offshore distance along major continents is about 50 km according to our ½° model resolution, with a few points showing larger distance due to specific rugged coastlines not properly accounted for in our relatively coarse resolution model mask."

Satellite data are also flagged out when too close to the coast before performing the co-location and comparison. This is also stated in the Methods section:

"Coastal values in a 50km along-shore area are flagged out because of the poor reliability of the data due to land interference with the signal. Inaccurately measured values are also flagged out based on the noise of the retrieved signals. In order to perform a proper comparison, both model outputs and along-track satellite Hs are extracted in a 0.5°x0.5° box on our coastal-extracted points (which are ~50km offshore)."

5. [Line 78] Consistent use of either WaveWatch-III or WWIII is recommended.

Corrected.

6. [Line 103] Should be written as "... sensitive to the percentile ...".

Corrected.

7. [Line 147] Suggested to rewrite as "... it can be noted that SAM exhibits a correlation coefficient of 0.43 with extreme wave ...".

Corrected.

8. [Line 233] Use of the word "prohibited" is unsuitable in the context. Suggested to rewrite as "The compulsion of using a coarse resolution arose grid was owing to the limitation of

high computational cost of the model running globally for a long duration (1993-2017).”, or in this lines.

Corrected.

9. [Line 234] Does the study period span the entire duration of 1993 – 2017, or only for the specific cyclone durations in this interval? The authors should add a clear statement in the text describing the study period.

Both simulations, with and without TCs, were conducted over the entire duration of 1990-2017. This is now clearly stated: *“Two WAVEWATCH III simulations are performed over the 1990-2017 period with two different forcing fields [...]”*

10. [Line 234] A justification for the choice of spectral space resolution and frequencies is suggested to be added here to introduce more clarity and foundation to the study.

The spectral discretization is also a compromise between precision and affordable computational cost. The directional discretization is 15° . Increasing to 10° or 5° could be interesting but was not affordable for our global study. We however use a 3rd order advection scheme for both spatial and spectral advection, and the correction of the garden sprinkler effect proposed by Tolman (2002), to limit the impact of the directional discretization. The frequency discretization, 0.037 to 0.71 Hz, allows to solve the main swell peaks and wind-induced waves. The high-frequency extent of the prognostic region is actually scaled according to the mean frequency of the local windsea. A diagnostic tail following a f^{-5} power law (e.g., WAMDIG, 1988) is added at higher frequencies to compute non-linear transfer in the prognostic region and to compute the integral quantities which occur in the dissipation source function and the wave induced stress. Alday et al. (2021) proposed a slightly extended frequency range up to 0.95Hz to improve the case of very low wind speeds or very short fetches, which are not the main targets of the present study. On the low-frequency side of the spectrum, we do not solve infra-gravity (IG) waves, but we use the parameterization of Ardhuin et al. (2014) for free IG waves, as they are important for very energetic events such as those we are looking at. This parameterization was shown to work fairly well for 30-200s IG waves, capturing between 30% and 80% of the variance in IG wave heights, and 50% of the mean IG energy.

Overall, the significant wave height and the wave energy flux both show very good agreement compared to satellite and buoy observations respectively (see new Fig. 1c and S1b), giving us confidence in the model ability to correctly simulate and propagate towards coastal zones the extreme wave energy fluxes.

The Methods section has been modified as follows:

“The spectral space is discretized in 24 directions (15° resolution), and 32 frequencies ranging from 0.0373 to 0.7159Hz with an increment factor of 1.1. The frequency range allows to solve the main swell peaks and wind-induced waves. A diagnostic tail following a f^{-5} power law [50] is added at higher frequencies. Beyond the lower frequency, we use the parameterization of [51], which has been shown to effectively represent the characteristics of free infra-gravity (IG) waves, particularly relevant for highly energetic events like those examined in our study. The ultimate quickest third order propagation scheme is used for both spatial and spectral advection, with the garden sprinkler effect correction proposed by [52], limiting the impact of the directional discretization.”

[Alday et al., 2021] <https://doi.org/10.1016/j.ocemod.2021.101848>

11. [Line 235] Simulation time-steps should be clearly mentioned somewhere in the text.

A sentence specifying the time steps has been added to the description of the model configuration: *“WAVEWATCH III uses a fractional time step method with 4 time steps. The global time step treating temporal variations of the depth is set to 2400s, the time step for spatial propagation is 480s, the time step for refraction is 300s, and the minimum time step for integrating the source terms is 10s.”*

12. [Line 239] Use of vague statements, such as “... mostly similar to ...”, are not recommended. Suggested to describe the parameterization schemes briefly.

This has been rephrased. The parameterizations are now fully detailed in the Methods.

“Beyond the lower frequency, we use the parameterization of [51], which has been shown to effectively represent the characteristics of free infra-gravity (IG) waves, particularly relevant for highly energetic events like those examined in our study. The ultimate quickest third order propagation scheme is used for both spatial and spectral advection, with the garden sprinkler effect correction proposed by [52], limiting the impact of the directional discretization. Non-linear wave-wave interactions are modelled using the discrete interaction approximation of [53]. The bathymetry used is GEBCO 30” in its 2014 version (http://www.gebco.net/data_and_products/gridded_bathymetry_data/). Sub-grid scale islands are addressed through an obstruction parameterization, which effectively reduces the wave energy of the cell in proportion to the obstruction [54-55]. Reflection at the coast is accounted for with a coefficient of 0.1 and with a dependence to frequency [56]. SHOWEX bottom friction [57] and depth-induced breaking parameterization [58] with a Miche-style shallow water limiter for maximum energy are used. For wave growth and wave dissipation the ST4 source term package [59] is chosen, with the wind-wave growth parameter, β_{max} , adjusted to 1.6, the sheltering coefficient set to 0.3, and the swell dissipation parameters set to SWELLF=0.69, SWELLF3=0.022, SWELLF4=150000, SWELLF7=468000.”

13. [Line 244] Either proper justification or citation is required to support the statement.

Reference added.

14. [Lines 245 – 247] Suggested to rewrite the sentence.

Rephrased, as detailed in reply to comment #12.

15. [Line 252] A list of the atmospheric variables used as forcings to the model will make the manuscript more informative.

The atmospheric forcing is performed with 10-m winds. The 2 different wind forcings used in the study are fully detailed in section “Experiments”.

16. [Lines 254 – 258] The sentence is too long and incoherent. Suggested to use shorter, clear sentences.

Corrected as suggested:

“This reanalysis features a reduced negative bias in wind speed compared to previous reanalyses [61], and was shown to improve TC storm surge simulations. However, a significant bias still remains for intense, small size, and fast moving TCs. The overall bias in

cyclone maximum wind speeds provided in the ERA5 reanalysis is -37% compared to the International Best Track Archive observations (see Supplementary material)."

17. [Line 262] What is the justification of filtering TCs by using 11-day running-mean technique, when more sophisticated methods of removing TCs and calculating background circulation exist? Have the authors looked into calculating background winds using the stream function and velocity potential [1] or using the weighted mean of winds through the atmosphere between 850 hPa and 250 hPa levels [2]? How different will the inferences of this study be if the authors would use either of these two methods?

[1] Arakane, S., Hsu, HH. A tropical cyclone removal technique based on potential vorticity inversion to better quantify tropical cyclone contribution to the background circulation. *Clim Dyn* 54, 3201–3226 (2020). <https://doi.org/10.1007/s00382-020-05165-x>

[2] Emanuel, K., Ravela, S., Vivant, E., & Risi, C. (2006). A statistical deterministic approach to hurricane risk assessment. *Bulletin of the American Meteorological Society*, 87(3), 299-314. <https://doi.org/10.1175/BAMS-87-3-299>

Indeed, the 11-day running-mean is straightforward technique, yet it has demonstrated effectiveness in generating surface wind forcing for ocean models (e.g. Vincent et al. 2012). While more sophisticated methods may be warranted for evaluating the impact of TCs on atmospheric circulation, as demonstrated in Arakane et al. (2020), we believe that within the scope of our study, employing such advanced techniques would not significantly alter the contribution of TCs to extreme waves.

[Vincent et al., 2012] <https://doi.org/10.1029/2011JC007396>

18. [Line 263] What is the justification of using a linear transition of winds between 600 and 1200 kms away from the cyclone tracks instead of using the actual observations?

The linear transition facilitates the creation of a wind field that smoothly transitions from the prescribed parametric TC vortex and the ERA5 background wind field, avoiding spurious signals. While observations of the TC periphery can be obtained through satellite scatterometers or radiometers, they are not continuous and were not available for all events throughout the entire duration of our simulation (1990-2017).

Justification and details about the linear transition have been added:

"To avoid a rough transition at 600km between the new wind field including the TC winds (referred to as $wind_{new}$ in the following) and the ERA5 native wind field, a linear transition was applied between 600 and 1200km, such as $wind = (1-\alpha)wind_{new} + \alpha \cdot wind_{ERA5}$, with α being a linear function ranging from 0 at 600km from the TC track to 1 at 1200km. ERA5 wind field remains untouched off 1200km of the TC tracks."

19. [Line 284] Is the subsampling of coastal points random, or have the authors followed any specific criteria to subset the coastal points?

The sub-sampling was performed according to the size of coastal polygons to keep small islands but also visual clarity. The procedure is now detailed in the method section:

"The simulation results are depicted along global coastlines and islands as defined by [67], albeit sub-sampled to enhance visual clarity and accommodate the relatively coarse resolution of our model. Sub-sampling is performed as follows: the coastline data [67] are first re-gridded on a $1/24^\circ$ grid, followed by sub-sampling of the coastline based on the size of each coastal polygon. For large polygons (more than 60 points), every 6 points (~25km) are retained, while for medium-sized polygons (between 15 and 60 points), every 3 points (~13km) are retained. All locations (~4km) are preserved for small polygons (less than 15

points). Subsequently, the resulting coastline undergoes meticulous visual inspection to eliminate any undesired closed contours that may have appear in rugged coastline areas. Finally, wave fields are extracted at the nearest deep-water location for each of these coastal points.”

Additional sub-sampling was performed for plotting pie-charts or bars by averaging the pie-charts or bars every 200 coastal points. This is now detailed in the figure captions.

20. [Lines 297 – 300] It is suggested that the authors rewrite this part to bring in more clarity.

This has been re-written as follows:

The wave energy flux (WEF), quantifying the amount of energy carried by ocean waves as they propagate, is computed as:

$$WEF = EC_g$$

with $E = \frac{1}{2}\rho g a^2$ the wave energy, a being the wave amplitude, $\rho = 1025\text{kg.m}^{-3}$ the water density, $g = 9.81\text{ m.s}^{-2}$ the gravitational acceleration, and C_g the wave group velocity. When waves propagate, and when they approach the shore, the flux rather than the energy, is conserved, making WEF a more relevant variable than H_s to assess wave coastal incident conditions. For an easier comparison of H_s and WEF at our coastal points, we use a proxy of WEF called the breaking wave height, H_b . It is computed assuming that wave breaks at a depth equal to the wave height [68], and considering that WEF is conserved from deep water to the depth of breaking:

$$WEF_{offshore} = WEF_b$$

with $WEF_{offshore}$ the WEF of deep water offshore condition, computed with $C_g = \frac{g}{4\pi} T_p$ the group velocity in deep water, and $a = \frac{H_s}{2\sqrt{2}}$ the wave amplitude in deep water:

$$WEF_{offshore} = EC_g = \frac{\rho g^2}{64\pi} H_s^2 T_p$$

and WEF_b the WEF at the depth of breaking, computed with $C_g = \sqrt{gH_b}$ the group velocity in shallow water, and $a = \frac{H_b}{2\sqrt{2}}$ the wave amplitude at the breaking depth:

$$WEF_b = EC_g = \frac{\rho g^{3/2}}{16} H_b^{5/2}$$

Therefore, H_b writes:

$$H_b = \left(\frac{H_s^2 \sqrt{g}}{4\pi} T_p \right)^{\frac{2}{5}}$$

21. [Lines 303 – 304] On what basis have the authors chosen 98th percentile as the threshold for extreme events?

There is no universally accepted guideline for selecting a threshold for extreme values: rather, it typically hinges on the specific impact the user aims to investigate. In this study, we opted for the 98th percentile. However, we conducted a sensitivity analysis on this threshold and included the corresponding TC contribution values for the 95th or 99.5th percentiles in the text. Additionally, we have provided maps with these alternative thresholds in supplementary material (Fig. S2).

22. [Line 317] It is known that the parameters behind NAO, SAM and Nino3.4 have intraseasonal to yearly oscillations. What is the justification of comparing yearly averages of these indices with contribution of TCs which are highly seasonal?

Since our focus is to examine extreme events occurrence (e.g. number of events above the 98th percentile) and their relation with these 3 modes of variability (Fig. 3a), we decided to consider all the events over a year.

As shown below, restricting the period to the peak period of ENSO and NAO (November to April) leads to similar results:

	ENSO yearly	ENSO peak	NAO yearly	NAO peak
TCs	0.57 (0.002)	0.65 (0.0004)	0.19 (0.35)	0.12 (0.58)
CYCL	0.53 (0.006)	0.58 (0.003)	0.03 (0.89)	-0.14 (0.51)
NOCYCL	0.11 (0.59)	0.09 (0.45)	-0.16 (0.45)	-0.32 (0.11)

For SAM, oscillations occur more frequently, leading us to aggregate data over one year to more accurately capture its influence .

For sake of simplicity and consistency, we decided to maintain the yearly period for all indices.

23. [Line 321] Is the criteria that Nino3.4 index should be more than 0.4 (less than -0.4) for at least six consecutive months followed in the calculation of ENSO phases? If yes, it may be mentioned in the text.

As detailed in response to comment #22, we considered yearly values, which actually captures the Niño/Niña events as the usual definition (5-consecutive months) does.

24. [Line 325] The line may be rewritten as "... number of respective Niño and Niña phases."

Corrected.

25. [Line 327] The term CCI should be expanded at first occurrence.

Corrected.

26. [Line 331] The line may be rewritten as "Inaccurately measured values ...".

Corrected.

27. [Line 534] On what basis have the authors chosen the 0.3 m threshold while defining the TC impact?

To define TC-impacted locations, we have selected areas showing difference in Hs 98th percentile between CYCL and NOCYCL simulations. The choice of the threshold being somewhat arbitrary, its sensitivity is further illustrated in Figure R1. Noteworthy, the simulation is shown to agree well with observations regardless the chosen threshold. Therefore, the conclusions drawn regarding this aspect are not significantly affected by the choice of threshold. The higher the threshold, the lower the number of points considered in the analysis. We have thus finally decided to select the 0.1m difference threshold for Fig. 1 inserted

in the manuscript. When considering the strongest events only, characterized here with a Hs 98th percentile changing by more than 1m when accounting for TCs (grey curve in Fig. R1), the simulation slightly overestimates extreme Hs. That is discussed in the Discussion section.

Fig. R1: Simulated versus altimeter 98th percentile of Hs at the locations affected by TCs. These locations are extracted where the difference between CYCL and NOCYCL for the 98th percentile of Hs reaches a given threshold defined for each curve in the legend. Dots represent the binned median of locations for Hs ranging from 1 to 10m.

28. [Line 538] Inconsistencies in the choice of study period (1990 – 2017 or 1993 – 2017) must be resolved.

The period of simulation is 1990-2017. But the validation was only possible over 1992-2017, as the satellite dataset starts in 1992. This has been detailed and corrected in the manuscript.

29. [Line 544] How did the authors calculate the cyclone density difference between Niño and Niña phases?

A probability density function (PDF) is used to illustrate the spatial distribution of TC occurrence. The PDF is computed as the normalized sum of anisotropic Gaussian functions for each TC occurrence, with standard deviations in the meridional and zonal directions respectively of 1.5° and 3°, as in [Jourdain et al. 2011, Jullien et al. 2014].

The cyclone density difference between El Niño and La Niña phases is computed as the difference between TC densities summed over Niño and Niña phases, and weighted by the total number of respective phases.

This has been added to the Methods section.

[Jourdain et al. 2011] <https://doi.org/10.1175/2010JCLI3559.1>

[Jullien et al. 2014] <https://doi.org/10.1007/s00382-014-2096-6>

30. [Line 549] The line should be written as “... events with the Niño3.4 ...”.

Corrected.

Reply to Reviewer #3 comments.

Reviewer #3 (Remarks to the Author):

This manuscript studies extreme waves generated by tropical cyclones. They claimed by designing twin oceanic wave simulations with and without realistic TC wind forcing from ERA5 (with filtered 2D parametric wind), they enable to quantify TC's role on extreme waves for the first time. They found out waves substantially contribute to extreme breaking heights in tropical regions (35-50% on average), reaching 100% in high-density TC areas like the North Pacific. TCs also impact remote TC-free regions, such as the equatorial Pacific experiencing on average 30% of its extreme wave events due to TCs.

However, the claimed novelty of the paper is not clear. The following paper "Seamless projection of global storm surge and ocean waves under a warming climate" modeled waves with WAVEWATCH III (the model used in this study) globally. They also considered the future projection. So it is not new to model waves globally. The other novelty the authors claim is replacing the TC wind fields in reanalysis with parametric wind field. It is well known that reanalysis cannot well resolve TCs, and previous studies (included one cited by the authors) has done replacing TC wind fields in reanalysis as in this research. It is also well known that TCs dominate extreme levels in high-density TC areas. Maybe the finding that tropical regions relatively far away from TC areas also show TC induced waves. However, the authors didn't explain why. In fact, this analysis focus on waves in deep ocean, 50 km away from the coast; it is not clear if these remote effect of TCs will reach the coastal areas. One thing the authors can do is simple check coastal storm surge and wave records and see if remote (with "remote" to be clearly defined) coastal areas are affected by TCs.

We appreciate the reviewer's insightful comments on our manuscript. As correctly pointed out, some studies did already use global-scale wave modeling, while others have made advancements in improving TC representation in reanalysis, as acknowledged in our introduction. Therefore, we do not assert that the originality of our work lies in the methodological developments (i.e. using global wave modeling or improving TC representation in reanalysis), but rather in its scientific objectives: specifically, quantifying the TC and non-TC contribution to extreme waves approaching the coastal areas, and examining how this contribution is influenced by major climate modes of variability.

The paper, quoted by the reviewer (Shimura et al., 2022) has a quite different focus. It aims at addressing the fate of Hs and surge in the future compared to present day. Owing to the climate model framework they use, their TC variability (both seasonal and interannual), over the present period, appears significantly different from the observations. In contrast, our methodology ensures keeping the observed TC distributions and variability, and proved to provide very well validated extreme waves approaching the coasts. We therefore achieve to quantify the contribution of TCs to wave extremes compared to other remote extreme swells, and to examine the influence of the main interannual climate modes.

This quantitative assessment is crucial for regions with limited availability of observation time series for incoming waves, such as vulnerable islands and territories, and holds significant implications for their risk management strategies.

There are also many confusions. The authors keep saying coastal impact and storms surge and wave setup (in motivation and implication) but they only studied the waves in deep ocean, which is far from the coast. In the discussion section, it is mentioned that “Numerous prior studies have highlighted the shortcomings in storm surge assessments, in relation with the poor representation of TC winds and extreme waves in global models [9-11][13- 14][17][22][38].” How is the connection of the wave study that has been done here with storm surge? Storm surge is the rise in sea level along the coastline. How can these waves impact the storm surge? Are they studying deep-water waves, which could differ from the effects of nearshore waves?

In many areas, wave setup can be the dominant term in the sea-level anomaly at the coast (e.g. Vousdoukas et al. 2016, Rueda et al. 2017). Our primary objective is to assess the incident extreme waves approaching coastal areas. This serves as an essential initial step in assessing coastal vulnerability. Prior to considering the complexity and specificities of the coastal area, it is imperative to address two key issues at global scale: (1) the representation of TCs in global models, which lack the resolution to realistically represent the TC wind fields and (2) the contribution of waves to the extreme water levels, a factor which is frequently overlooked. Our study directly tackles these challenges to provide reliable extreme offshore wave conditions. These developments offer a valuable resource for evaluating wave-induced setup at a global scale along coastlines in subsequent dedicated studies.

Transitioning from offshore wave characteristics to actual storm inundation on land requires assessing wave transformation at the coast and in shallow waters environments, which is strongly site dependent. This is beyond the scope of our paper, and this is precisely why we have considered deep-water incident wave conditions, hence avoiding any hypothesis on coastal wave transformation.

We have revised the introduction and discussion to clarify our objectives and the implications of our study. We have also removed occurrences of the misleading ‘vulnerability’ notion along the text, and modified the title to: “*Unveiling the Global Influence of Tropical Cyclone on Extreme Waves approaching coastal areas*”.

The motivation started with tropical and extra-tropical cyclones, but didn’t continue to discuss about extra-tropical cyclone waves.

Extra-tropical storm, being the most important source of extreme waves besides TCs, are actually considered in the “non-TC” events. They are evaluated through the NOCYCL simulation (Table 1, new Fig. 3c, Fig. 4b), and their contribution is evaluated as the counterpart of TC contribution.

We have now more clearly highlighted the extra-tropical storm wave influence by changing the title of the interannual variability section to: “*Interannual variability of TC and non-TC wave hazards*”, and discussing it in more details in this section. Panel c has been added in Figure 3, according to another reviewer suggestion, to illustrate the non-TC events variability with ENSO phases.

Please explain the physical meaning of wave energy flux and why it is an important metric for what.

The wave energy flux (WEF) quantifies the amount of energy carried by ocean waves as they propagate. It is computed as:

$$\text{WEF} = E \cdot c_g$$

with E the wave energy and c_g the wave group velocity.

When waves propagate and approach the shore, the flux, rather than the energy, is conserved (Whitham 1962). This is why WEF is a more relevant variable than the significant wave height to assess wave coastal incident conditions. This is now clearly described in the Methods section.

Not clear how they modeled the wind field. Historical TC storm track and intensity information are available globally. Why do you need to use Emanuel database for observed maximum wind? Storm size information is only available for Atlantic, so how do you estimate size for other regions (L205 needs explanation and reference. L213, “overestimate TC size” by which model?). Given the storm parameters, how do you model the wind field?

The Emanuel database is based on the international best-track (IBTRACS) database, but with some corrections to the wind speed to compensate for changes over years and centers in the methods for estimating 1-min sustained winds (see the product documentation: <https://texmex.mit.edu/ftp/pub/emanuel/HURR/tracks/>). The wind field is constructed by blending ERA5 winds and a parametric TC formulation using the observed TC location and maximum wind speed data from Emanuel’s database. Because the radius of maximum winds is not available for all basins, it is parameterized from the maximum wind speed and the latitude (see Willoughby et al., 2006). This has been clarified in the Methods section:

“To better represent the effects of TCs, we replaced the ERA5 surface wind in TCs by a parametric structure reconstructed from the observed maximum wind speed following a procedure similar to that of [62]. We first filtered out the weaker than observed TCs in ERA5 with an 11-day time running-mean filter within 600-km of each cyclone track position. Then a 2D surface parametric wind field, reconstructed using the observed maximum wind speed in TCs provided by K. Emanuel database (<ftp://texmex.mit.edu/pub/emanuel/HURR/tracks/>) and the parametric model of [44], is added to the filtered wind field at each cyclone position between 0 and 600km from the track. The parametric model was calculated so that the amplitude reached at the TC center in the reconstructed forcing was equal to the observed intensity, and using a radius of maximum TC winds ideally derived as a function of latitude and maximum wind [44]. To avoid a rough transition at 600km between the new wind field including the TC winds (referred to as $wind_{new}$ in the following) and the ERA5 native wind field, a linear transition is applied between 600 and 1200km, such as $wind = (1-\alpha)wind_{new} + \alpha \cdot wind_{ERA5}$, with α being a linear function ranging from 0 at 600km from the TC track to 1 at 1200km. ERA5 wind field remains untouched off 1200km of the cyclone tracks. Such procedure has been shown to be very successful in reproducing the observed ocean hydrodynamics under TCs [62]. Two WAVEWATCH III simulations are performed over the 1990-2017 period with two different forcing fields: the ERA5 surface winds merged with parametric TCs of observed amplitudes (referred to as “CYCL”), and the ERA5 surface winds with filtered TCs (referred to as “NOCYCL”). Difference between simulations CYCL and NOCYCL is used to isolate the impact of TCs on coastal extreme sea states. NOCYCL simulation is used to assess the impact of non-TC events.”

It is well known that climate state represented by index like ENSO affects TCs and thus TC induced waves. What is the implication of the results in this component?

Yes, it is well known that ENSO phases have strong impact on the TC densities (location and number of TCs). Our study provides a quantification, over a long period of time, of the modulation of extreme waves approaching the coasts in response to ENSO. Our results also show that the main driver of the TC extreme wave changes is indeed the change in TC densities. This is discussed in the associated section:

“Extreme wave events exhibit a significant correlation with ENSO (0.53, p-value=0.006), especially those associated with TCs (significant correlation of 0.57, increasing to 0.65 when considering the ENSO peak period, November to April, only). During El Niño phases, the Central Pacific faces a significantly higher threat from wave extremes (Fig. 3a), with twice as many events compared to La Niña or Neutral phases. Conversely, regions such as the South China Sea, the Indonesian archipelago, the tropical Atlantic basin, or the northern tip of the Pacific experience an increase in extreme wave events during La Niña phases (Fig. 3a). Changes in TC densities associated to ENSO largely drives the spatial pattern of this interannual variability in coastal wave extremes (Fig. 3b). The area of TC occurrence is shifted eastward in the North-Western Pacific, eastward and equatorward in the South-West Pacific in response to the South Pacific Convergence Zone shift during El Niño [36-37], increasing the exposure of the Micronesian, Melanesian, Polynesian, and Central Pacific Archipelagos. These results align with previous regional studies by [32] and [33]. During La Niña phases, TC densities and associated extreme waves increase along the North West coasts of Australia, in the Mozambique channel, in the South China Sea, and in the Caribbean region. Wave extremes of non-TC origin are although also modulated by ENSO (Fig. 3c), with an increase of Southern swells propagating across the Pacific basin observed during El Niño phases [38], and an increased number of extreme waves in the Northern Pacific, Eastern Atlantic, and along the Indonesian coasts during La Niña.”

There are numerous language errors. Some examples:

L38, “wave setup, i.e., ..., is often lacking.

Rephrased.

“Secondly, the contribution of waves to extreme water level, i.e. the wave setup, is often overlooked [...].”

L54-57, quoting a very long and confusing sentence from others

Removed. The introduction has been re-written to highlight gaps in the literature, and better justifying the objectives of the study.

L70-73, Objectives of the study are very confusing (and about coastal and coastlines). “assessing the contribution of TCs compared to remotely generated swells...” (contribution of TCs compared to contributions of remotely generated swells???)

Corrected.

“Providing reliable extreme offshore wave conditions which will serve as a valuable resources for... (how do you provide wave conditions?)

A public link and DOI for simulated wave fields, extracted at the nearest deep-water location for each of the coastal points, used in the study will be provided (data are currently in the process of being stored in a public repository).

L116-121. Note “vulnerability” is related to the system or coastline affected by the waves, it’s the inability of the system to withstand the hazards. “wave-induced vulnerability” is very confusing.

Yes, sorry for this language error. Changed to “hazards”.

L245-246 “[56]parameterization..., ..., ..., is...”

The sentence has been rephrased, and details added in response to another reviewer comment:

“The parameterization used for wave growth and wave dissipation is the ST4 source term package [56], in which the wind-wave growth parameter, β_{max} , is adjusted to 1.6, the sheltering coefficient is set to 0.3, and the swell dissipation parameters are set to SWELLF=0.69, SWELLF3=0.022, SWELLF4=150000, SWELLF7=468000.”

L267. Please explain the linear transition.

The linear transition facilitates the creation of a wind field that smoothly transitions from the prescribed parametric TC vortex and the ERA5 background wind field, avoiding spurious signals. Details have been added to the Method section:

“To avoid a rough transition at 600km between the new wind field including the TC winds (referred to as $wind_{new}$ in the following) and the ERA5 native wind field, a linear transition is applied between 600 and 1200km, such as $wind = (1-\alpha)wind_{new} + \alpha.wind_{ERA5}$, with α being a linear function ranging from 0 at 600km from the TC track to 1 at 1200km. ERA5 wind field remains untouched off 1200km of the cyclone tracks.”

L278. “As the knowledge of ...and evolves in time,...” (knowledge evolves in time???)

We have rephrased the sentence:

“As the shoreline morphology is poorly known in numerous locations and additionally may evolve in time (erosion, deposition), the assessment of the contribution of waves to the water level at a global scale is thus very difficult [...].”

REVIEWER COMMENTS

Reviewer #1 (Remarks to the Author):

The authors have addressed my comments. I recommend this paper be published.

Reviewer #2 (Remarks to the Author):

All the concerns raised in the review are suitably addressed by the authors in the revised version.
D. Swain

Reviewer #3 (Remarks to the Author):

Thanks for responding to the comments. The introduction and conclusion now effectively highlight the novelty of the paper, which investigates the contribution of TCs to ocean wave flux, comparing long-term wave hindcast with and without influences of TCs. One of the remaining concerns is the lack of validation for the wave energy flux, as the validation primary focused on the significant wave height. According to the authors' response, the significant wave height is not the central focus of the study, then should the wave energy flux be validated? Please clarify.

Reply to Reviewer #3 comments.

Reviewer #3 (Remarks to the Author):

Thanks for responding to the comments. The introduction and conclusion now effectively highlight the novelty of the paper, which investigates the contribution of TCs to ocean wave flux, comparing long-term wave hindcast with and without influences of TCs. One of the remaining concerns is the lack of validation for the wave energy flux, as the validation primary focused on the significant wave height. According to the authors' response, the significant wave height is not the central focus of the study, then should the wave energy flux be validated? Please clarify.

A validation of the breaking wave height (H_b), which serves as proxy for the wave energy flux, was added in the previous revised version of the manuscript, according to another reviewer comment. Fig. 1c, along with its description, was added to the manuscript, as well as Fig. S1b, which was added to the supplementary material.

H_b is validated using data from available buoys, primarily sourced from NDBC. These buoys were carefully selected to be positioned at least 50km offshore and in water depths exceeding 50m, aligning with the deep-water criterion of our study. Throughout the period 2012-2016, approximately 60 to 80 buoys measuring H_s and T_p were available each year. Our analysis reveals a very good agreement between the model and buoy data for the 98th percentile of H_b (Fig. 1c of the manuscript). CYCL simulation exhibits a normalized bias of 0.6%, a correlation of 0.96 and a normalized root mean square error of 8.7%. When focusing solely on locations affected by TCs, the normalized bias is slightly positive but remains weak in our CYCL simulation (4.3%). Larger biases arise in the NOCYCL simulation (-6.0% and -8.6%) for respectively all and TC-affected locations. These results underscore the model's ability to accurately simulate the extreme wave energy fluxes propagating towards coastal zones.